# Direct-to-consumer tests advertised online in Australia and their implications for medical overuse: systematic online review and a typology of clinical utility

Patti Shih ,[1] Pauline Ding,[2] Stacy M Carter,[1] Fiona Stanaway ,[3] Andrea R Horvath,[4,5] Daman Langguth,[6] Mirette Saad,[7] Andrew St John ,[8] Katy Bell [3]

For numbered affiliations see end of article.

**Correspondence to**
Dr Patti Shih;
pshih@uow.edu.au

## ABSTRACT

**Objectives** The objective of this study is to map the range and variety of direct-to-consumer (DTC) tests advertised online in Australia and analyse their potential clinical utility and implications for medical overuse.

**Design** Systematic online search of DTC test products in Google and Google Shopping. DTC test advertisements data were collected and analysed to develop a typology of potential clinical utility of the tests at population level, assessing their potential benefits and harms using available evidence, informed by concepts of medical overuse.

**Results** We identified 484 DTC tests (103 unique products), ranging from $A12.99 to $A1947 in cost (mean $A197.83; median $A148.50). Using our typology, we assigned the tests into one of four categories: tests with potential clinical utility (10.7%); tests with limited clinical utility (30.6%); non-evidence-based commercial 'health checks' (41.9%); and tests whose methods and/or target conditions are not recognised by the general medical community (16.7%). Of the products identified, 56% did not state that they offered pretest or post-test consultation, and 51% did not report analytical performance of the test or laboratory accreditation.

**Conclusions** This first-in-Australia study shows most DTC tests sold online have low potential clinical utility, with healthy consumers constituting the main target market. Harms may be caused by overdiagnosis, high rates of false positives and treatment decisions led by non-evidence-based tests, as well as financial costs of unnecessary and inappropriate testing. Regulatory mechanisms should demand a higher standard of evidence of clinical utility and efficacy for DTC tests. Better transparency and reporting of health outcomes, and the development of decision-support resources for consumers are needed.

## STRENGTHS AND LIMITATIONS OF THIS STUDY

⇒ This review strategy of direct-to-consumer (DTC) tests advertised online is comprehensive and further analysed with a conceptual focus on the clinical utility of DTC tests. It provides a nuanced and critical angle on their implications for medical overuse and potential benefits and harms for consumers, and can be replicated and applied outside of Australia.

⇒ The DTC tests included in this time-specific search may not be exhaustive, as product availability may change over time, and others may be advertised in different platforms or sold in pharmacies.

⇒ DTC tests offered by private commercial clinics but which are not advertised as a directly purchasable item online, as is often the case for genetic and fertility tests, were also not included.

⇒ This is an overview of products available; data extraction did not include an in-depth qualitative assessment of the textual information in each advertisement.

DTC testing does not require the recommendation or referral from a consulting doctor, nor the facilitation of a public health programme. This makes them distinct from home collection kits that are part of screening programmes (eg, bowel cancer screening), pandemic responses (eg, COVID-19 rapid antigen tests (RATs)), point-of-care testing (as this is conducted within a clinical or care-based setting) or self-monitoring tests for patients already diagnosed with a condition or under clinical management (eg, diabetes self-monitoring).

The size of the DTC test market is rapidly growing, expanding from a reported US$15.28 million per annum in 2010 to over US$352 million in 2020.[3] As part of a greater commercial shift towards more individualised

## INTRODUCTION

Direct-to-consumer (DTC) tests are commercially marketed pathology tests, purchased by members of the public,[1 2] who pay for and decide on which test to access and when.

and consumer-driven models of healthcare, DTC testing promises the opportunity for more proactive self-care and personal autonomy.[4 5] No longer just reliant on screening programmes or doctors' recommendations, consumers may (in theory) initiate and obtain pathology tests quickly and conveniently to inform early disease prevention or enable timely management decisions. Bypassing clinical visits may allow more privacy and confidentiality around testing, and may avoid the additional cost and logistics of travel and administration.[5]

Studies of DTC tests in the USA and UK show a wide variety of diagnostic, screening and risk monitoring tests being available for direct purchase, suggesting consumers with and without symptoms are targeted.[6–8] However, only 15% of the DTC screening tests found in the USA were supported by evidence-based recommendations,[6] and in the UK, only 1 out of 24 companies offering allergy testing used approved methods from an accredited laboratory.[8] These findings suggest the need to scrutinise the clinical utility of DTC tests: whether the product leads to useful clinical decision-making and improved health outcomes. Clinical utility depends on getting the right test for the right person at the right time.[9] This is determined by a variety of factors, such as an individual's medical history, symptoms (if any), and pretest probability for the target condition. Other important considerations are the reliability and validity of the testing method, and the quality of the in vitro device and the credibility of the laboratory. When products are directly marketed, consumers may be making test decisions without the adequate opportunity for discussion and guidance from their doctors.[10] Indeed, several studies show that the quality and reliability of information on websites promoting tests varies,[6 11 12] which can expose consumers to the harms of choosing inappropriate and/or unnecessary tests.

An often underanalysed harm related to pathology tests of limited clinical utility is medical overuse: healthcare that is either harmful or does not lead to improved health outcomes.[13 14] This includes using tests inappropriately or unnecessarily, such as testing low-risk and asymptomatic populations outside of clinical recommendations. This can drive the unnecessary diagnosis of early or slow-progressing diseases that would not have harmed the patient in their lifetime (overdiagnosis), and instead lead to treatments that produce no significant benefit but may cause harm (overtreatment).[13 14] Testing, whether using home testing kits or at a laboratory, is resource intensive and produces medical waste.[15] Testing demands further clinical resources such as analysis and interpretation, and decision-making based on results, whether positive or negative.[16 17] Test results can also trigger downstream testing and unjustified treatments that generate further costs and resource-use in the health system, such as unnecessary general practitioner (GP) visits.[18 19] When testing decisions are not clinically justified or do not lead to improvement in health outcomes, they are a form of 'low-value care', creating waste and additional burden to the healthcare system.[20] Although it is well recognised

that medical overuse occurs even in clinician-guided settings,[13 21] it may be exacerbated in a DTC context where consumers may be making testing decisions with little support or balanced information.[17]

Considerable critical attention has already been paid to the questionable clinical utility of the growing DTC genetic testing and fertility treatment industry.[22–24] For example, genetic tests for conditions such as Alzheimer's disease may predict a person's risk of eventually developing the condition where prevention strategies are limited, and testing can also cause psychosocial distress to healthy people.[25] The results of the anti-Mullerian hormone ('Egg Timer') test measuring ovarian reserve are unreliable as a basis for making family planning decisions in the future, as the test is only potentially useful for patients undergoing a current fertility treatment.[12 26] Typically, these better-known DTC genetic and fertility tests are offered by private clinics that include a suite of services and consultations before tests are chosen and performed 'in-house'.

An emerging trend in DTC testing that has had limited exploration is the 'direct purchase' online testing market: pathology test products and services advertised widely on the internet, and immediately purchasable in a single online transaction. Broadly, there are three types of directly purchased DTC tests in Australia: (1) Home self-tests: test kits used by consumers themselves producing immediate results for their own interpretation; (2) Self-collected direct accessed pathology tests (DAPTs): a pathology test requested from a commercial company on behalf of a consumer who purchased the test, rather than requested by a doctor after consultation and discussion, with samples collected at home and mailed into a laboratory or (3) Lab-collected DAPTs, again purchased directly by a consumer, but with samples collected in a laboratory. Home self-tests and DAPTs signal different levels of involvement of laboratory professionals and have different regulatory requirements depending on the jurisdiction. For example, in Australia, home self-tests are regulated by the Therapeutic Goods Administration (TGA), whereas laboratory-tested DAPTs are regulated by the National Association of Testing Authorities (NATA). The commonality is that consumers are the primary decision-makers in testing and purchasing decisions. While some DTC testing services do offer professional consultations as an inclusion or as an add-on purchase, if this happens at all, testing decisions are already made before there is an opportunity to discuss and deliberate about the personal values and expectations of the consumer in relation to taking a specific test.[27]

Currently, there is limited research available on DTC tests available in Australia, nor has there been much assessment of the potential benefits and harms of the different products available. This first-in-Australia study reviews DTC tests advertised online, with a focus on their clinical utility and implications for medical overuse. The research questions for this study are: (1) How many and what variety of products are advertised and available to

**Table 1** Data search and selection strategy

| Search strategy | Description | |
|---|---|---|
| Search string 1 | Home self-tests and self-collected pathology tests | Search: "self test", "self diagnosis", "home test", "home diagnosis", "home screen", "self screen". |
| Search string 2 | Direct access pathology and self-collected pathology tests | Search string 2: "direct access test*", "self request* pathology", "private blood test*", "private medical test", "online blood test". |
| Search inclusions and exclusions | | The published region was set in Australia. The search excluded the term COVID, and .gov and.edu domains, to omit any screening tests that are offered by government screening programs. |
| Selection inclusion | To be included in the study:<br>For all tests:<br>1. The tests are intended to indicate the presence of disease or conditions (eg, diagnose a disease), or for general well-being purposes (eg, tests that purport to indicate level of hydration, vitamins, hormones)<br>2. The product can be shipped to an Australian address<br>3. A test must be a purchasable product, itemised and displayed in AUD that is able to be purchased via an online transaction<br>▶ Home self-tests: must include an in vitro device that collects a human tissue sample (eg, a lancet or container for blood, urine, saliva, faecal material); samples are processed at home (eg, testing strip)<br>▶ Self-collected DAPT: must include an in vitro device that collects a human tissue sample (eg, a lancet or container for blood, urine, saliva, faecal material); samples are sent to be processed at a laboratory<br>▶ Lab-collected DAPT: must be a pathology request for a named biomarker or analyte that is tested in a laboratory | |
| Selection exclusion | The product is excluded if it is:<br>1. Not for testing human samples (eg, animals, soil, water)<br>2. Intended for use by professionals or clinics, for example, doctors, hospitals or laboratories<br>3. Used or purchased with a separate product (eg, used with a metre or reader or as part of a health coaching programme)<br>4. Funded as part of a government or health group screening programme<br>5. For indicating the presence of illicit drugs or alcohol, and non-health related purposes (eg, paternity and ancestry)<br>6. For indicating pregnancy, monitoring a pre-existing condition, testing for SARS-CoV-2 infection | |

DAPT, direct accessed pathology test.

Australian consumers online? (2) What are their implications for medical overuse?

## METHODS

The study took place in two stages. First, a systematic search of DTC test products advertised online was conducted in Australia; Second, the data collected were developed into a typology of potential clinical utility of the product, which was analysed based on the potential benefits and harms of the test at a population level, informed by concepts of medical overuse.

### Stage 1: systematic online search

A pilot search was conducted in Google and Google Shopping by two researchers independently in October 2020. These engines were chosen because the majority of internet searches for product purchases in Australia are conducted on these platforms. The pilot search identified the three modes of DTC tests (home self-tests, self-collected DAPT and lab-collected DAPT) and a range of medical conditions and/or types of tests. This helped refine the search terms used in an updated search

(table 1), taking place between June and August 2021. We allowed snowball inclusion of new items between August and December 2021, whereby additional products that met the study inclusion criteria were added if they appeared on the same websites advertising already-included products. Google search engine algorithms display search results according to relevance ranking by numerical order: the first ranked results and pages will be the most relevant, with a decline in relevance of search results in subsequent pages. Despite the searches returning a large number of total results, only very small, negligible percentages of relevance are expected in later pages, particularly after 200 results.[28 29] Therefore, we reviewed the first 20 pages (10 results per page, totalling 200 results) from each search. While it can never be guaranteed that this cut-off point will not miss relevant results, it is the most practically and logistically viable approach to conduct the review with a standardised cut-off point for each search scan. We excluded sites that were not relevant, and any duplicate results. We created a data extraction table to chart data relevant for the assessment of clinical utility and medical overuse, and collected these

data from the website of each included product: name of product, condition and/or analyte tested for, specimen sampled, mode of the test, cost in Australian dollars (including shipping), whether the product offers any pretest or post-test consultations; and any indicators of the quality assurance of the tests (for home self-tests, this is indicated by information about the diagnostic accuracy of the test, expressed as clinical sensitivity and specificity metrics; and for self-collected DAPT and lab-collected DAPT, this is indicated by any information on the accreditation of nominated laboratories or individual tests).

## Stage 2: developing a typology of clinical utility

As an emerging new area of research, there were no available or comparable typologies of clinical utility that could be applied as a framework to the DTC context. We used an abductive analytical approach to make sense of the data, which combines both inductive and deductive data analysis.[30] Inductive approach allows key concepts to emerge from data itself; and the application of preexisting theories and knowledge deductively forms a more structured understanding of data.[30] The analysis was led by the first author with input from the multidisciplinary authorship team, which included experts in medical overuse (KB, SMC and ARH) and pathology (ARH, DL, MS and ASJ). The data were first viewed as a whole, and then grouped and sorted into broader categories based on the key similarities and distinctions over several iterations. The grouping and sorting process was informed by pre-existing concepts and our own prior knowledge of the medical overuse literature, especially in relation to the overuse of pathology tests. The following principles from the medical overuse literature most relevant for pathology tests were used to guide the analysis:

► The balance of potential harms and benefits of the test for the individual consumer is based on the potential harms and benefits using population-level data.
► The scientific and analytical validity of the test as appropriate for the condition and/or purpose stated is based on published literature supporting scientific evidence, selected using the hierarchy of evidence as determined by the Oxford Centre for Evidence-based Medicine.[31]
► We prioritised any information identified by existing recommendations for or against specific tests by professional organisations, especially those developed to promote greater test stewardship (eg, National Institute for Health and Care Excellence (NICE), Choosing Wisely).

We moved iteratively back and forth between data and theory to change or add new inclusion/exclusion criteria to the groupings until no new categories emerged, and there was consensus between the authors. The allocation of tests into each category was supported by a literature search of evidence-based medicine (EBM) (seeonline supplemental file 1) The final categories were determined by the following levels of distinctions (see online supplemental file 2): The first key distinction made was whether there is sufficient evidence base to support the validity of the test as a diagnostic or screening tool for the stated condition it tests for. Second, within the evidence-based tests, there was a further distinction between those with and without a professional recommendation for screening the condition among specific populations. In the non-evidence-based tests, there was a distinction between those that aimed to detect a disease and those that aimed to promote general health. Third, within the evidence-based tests with no recommendations for screening, tests that are 'recommended against' by known professional guidelines aimed at preventing medical overuse were highlighted, followed by tests that have other medical overuse concerns, for example, if the test method itself has low specificity or sensitivity, or that the condition being tested for has low pretest probability. Within the non-evidence-based tests for disease, a distinction was made between tests that were non-evidence based in themselves, and tests used to diagnose conditions that were not recognised by the general medical community. In total, four categories emerged, with two of the categories (2 and 4) having three and two subcategories, respectively.

## Patient and public involvement

Patients and the public will be involved in the dissemination plans of our research. The study results will be promoted publicly with health consumer partners and advisors from the Consumer Reference Group of the Wiser Healthcare, a National Health and Medical Research Council Centre for Research Excellence, the Consumers Health Forum of Australia, and Health Consumers New South Wales.

## FINDINGS

A total of 484 DTC tests were identified. Collectively these targeted 103 unique medical conditions, analytes or test products. Tests ranged from $A12.99 to $A1947 in cost (mean $A197.83; median $A148.50). There were 177 (36.6%) home self-tests, 65 (13%) self-collected DAPT, and 242 (51%) lab-collected DAPT.

## Categories of clinical utility

### Category 1: tests with potential clinical utility (10.7% of total)

Just over 1/10 (10.7%) of total products were allocated to category 1 (table 2). These were evidence-based tests recommended for screening in specific population groups by a recognised professional medical organisation, determined by a reported EBM approach. The most common examples were tests for sexually transmitted infections (STI), for which stigma contributes to barriers to clinical care, especially among communities at higher risk. In our study, DTC tests for STI tended to come in the form of home self-testing kits, which are lower in cost compared with DAPTs and more likely to report test accuracy. Studies in Australia and internationally show that undertested at-risk populations are more likely to use

**Table 2** Category 1 results, tests with potential clinical utility

| Inclusion criteria | Target condition or test purpose | Key analyte or testing method | Mode of test* | | | | Min. cost | Max. cost |
|---|---|---|---|---|---|---|---|---|
| | | | H | S | L | TN | | |
| An established and recognised part of the diagnostic, screening or risk assessment pathway for the named medical condition or risk factor of a condition the product purports to test for. Recommended for screening in specific population groups by a recognised professional medical organisation, determined by a reported evidence-based medicine approach. | Bowel cancer | Faecal occult blood | 0 | 1 | 1 | **2** | 69.00 | 129.00 |
| | Cardiovascular disease risk factors | Lipoprotein | 0 | 0 | 3 | **3** | 198.00 | 330.00 |
| | Chlamydia | *Chlamydia* trachomatis antigen | 4 | 0 | 0 | **4** | 24.95 | 55.29 |
| | Cholesterol level | Blood cholesterol | 1 | 1 | 1 | **3** | 45.00 | 84.00 |
| | Diabetes | Hemoglobin A1c, glucose | 0 | 2 | 4 | **6** | 35.00 | 78.00 |
| | Gonorrhoea | *Neisseria gonorrhoea* antigen | 3 | 0 | 0 | **3** | 34.95 | 56.71 |
| | Hepatitis B | Hepatitis B surface antigen | 4 | 0 | 0 | **4** | 24.95 | 56.71 |
| | Hepatitis C | Hepatitis C virus antibody | 2 | 0 | 0 | **2** | 56.71 | 92.40 |
| | HIV | HIV-1 and HIV-2 antibody | 11 | 0 | 0 | **11** | 34.95 | 138.00 |
| | Sexually-transmitted diseases - multiple tests | Chlamydia antigen, Gonorrhoea antigen, hepatitis B antigen, syphilis | 2 | 0 | 7 | **9** | 64.00 | 188.00 |
| | Syphilis | Treponemal antibody | 4 | 0 | 0 | **4** | 24.95 | 56.71 |
| | Trichomoniasis infection | Trichomonas vaginalis lipophosphoglycan antibody | 1 | 0 | 0 | **1** | 56.71 | |

*Mode of access.
DAPT, direct accessed pathology test; H, home self-tests; L, lab-collected DAPT; S, self-collected DAPT; TN, total number.

online and home-based testing for STI than at primary care settings.[32 33] DTC tests in this category may improve the rate of beneficial diagnosis and treatment in high-risk populations, especially those with barriers to clinical care. However, the tests will have limited utility for asymptomatic and low-risk populations outside of populations recommended for routine testing. And even for people in high-risk populations, clinical follow-up is still needed to ensure timely treatment. This is especially notable since category 1 tests are the least likely to offer pretest and post-test consultation.

### Category 2: tests with limited clinical utility (30.6%)

While tests in category 2 (table 3) have a legitimate role in diagnosis or treatment in clinical settings for the condition they target, the context in which these are marketed as DTC tests bring up several concerns for medical overuse. First, about half in this category (designated as subgroup 2A, 19.4% of total), were tests identified by well-known guidelines that aim to reduce medical overuse, such as the Choosing Wisely 'Do Not Do' recommendations and the, NICE 'Do Not Do' Recommendation, and 'Screening tests of unproven benefit' identified by the Royal Australasian College of General Practitioners 'Guidelines for preventative activities in general practice'.[34–36] Examples include prostate-specific antigen and MTHFR gene testing. Other tests in this category were more likely to generate high false positive rates because of the limited accuracy of the testing method for the target condition (subgroup 2B, 9.3% of total). Examples include urine dipsticks for kidney and liver disease, which are often used in point-of-care screening before further investigation; whereas

when used by consumers at home, false positive results may cause unnecessary distress and trigger further GP visits. Third, a small proportion were tests for conditions with low pretest probability in the general population[37] (subgroup 2C, 1.9%). This means more potential for producing high rates of false positives or misattributed diagnosis. For example, an IgG test for Cytomegalovirus is of limited utility because a high proportion of the general population will carry antibodies, but the pretest probability for an active infection causing clinically significant disease is usually low. Even though the test may correctly show positive results from past vaccination or infection, it has limited utility in diagnosis, especially if symptoms are vague or within normal limits.

### Category 3: non-evidence-based commercial 'health checks' (41.9%)

Category 3 (table 4) were non-evidence-based commercial 'health checks', making up the largest proportion of all tests in this study, at 41.9% of total. This finding suggests that healthy people form the largest target of the DTC test marketing. These tests may sometimes be justifiable if offered in a clinical setting as part of investigating symptoms or disease monitoring (eg, testosterone hormone tests for hypogonadism). However there is negligible evidence to support their clinical utility as evaluative tests for health status among healthy populations.[38] A health check could be potentially useful if clinically relevant abnormalities are able to be detected. However, the tests identified in table 4 are not used for investigating symptoms of a known medical condition. Furthermore, there are no agreed protocols or harmonised range to interpret whether the test results fall into a healthy

**Table 3** Category 2 results, tests with limited clinical utility

| Inclusion criteria | Target condition or test purpose | Key analyte or testing method | Mode of test* | | | | Min. cost | Max. cost |
|---|---|---|---|---|---|---|---|---|
| | | | H | S | L | TN | | |
| **Subgroup 2A (n=94, 19.4% of total)** | | | | | | | | |
| Identified as potentially contributing to medical overuse in certain populations by professionally recognised and evidence-based medicine recommendations that aim to prevent medical overuse (eg, Choosing Wisely, National Institute for Health and Care Excellence (NICE) 'Do Not Do' Recommendations) | Alzheimer's disease | APOE genotype | 0 | 1 | 1 | 2 | 165.00 | 270.00 |
| | Arthritis | Rheumatoid Factor | 0 | 0 | 2 | 2 | 44.00 | 130.00 |
| | Autoimmune disease | HLA-B27 antibody | 0 | 0 | 2 | 2 | 69.00 | 120.00 |
| | Cancer markers | Carcinoembryonic antigen | 0 | 0 | 3 | 3 | 80.00 | 160.00 |
| | Cancer risk factors | Glutathione S-transferases genes | 0 | 0 | 2 | 2 | 313.00 | 313.00 |
| | Cardiovascular disease risk factors | Homocysteine | 0 | 0 | 1 | 1 | 747.00 | |
| | Cardiovascular disease risk factors | MTHFR and APOE genetic test | 0 | 1 | 0 | 1 | 199.00 | |
| | Coeliac disease | HLA gene typing | 0 | 0 | 1 | 1 | 169.00 | |
| | Food allergy | IgE | 0 | 2 | 5 | 7 | 90.00 | 430.00 |
| | Genital herpes | Antibodies | 3 | 0 | 0 | 3 | 63.00 | 140.40 |
| | Haemochromatosis | HFE gene mutation | 0 | 0 | 2 | 2 | 69.00 | 159.00 |
| | Heavy metal exposure | Heavy metal in blood | 0 | 0 | 3 | 3 | 118.00 | 189.00 |
| | Immune disorders | Immunogenetics | 0 | 1 | 0 | 1 | 440.00 | |
| | Insulin resistance | Fasting glucose, fasting Insulin | 0 | 0 | 2 | 2 | 69.00 | 190.00 |
| | Iron deficiency | Iron studies | 0 | 1 | 8 | 9 | 58.00 | 318.00 |
| | Low-grade Inflammation | High sensitivity CRP | 0 | 0 | 1 | 1 | 142.00 | |
| | MTHFR gene mutation | MTHFR gene mutation | 0 | 4 | 5 | 9 | 50.00 | 520.00 |
| | Neuro genetic disorders | Neurogenetics | 0 | 1 | 0 | 1 | 647.00 | |
| | Prostate cancer | Prostate-specific antigen | 0 | 0 | 2 | 2 | 39.00 | 40.00 |
| | Thyroid disease | Thyroid hormones | 1 | 7 | 15 | 23 | 35.00 | 328.00 |
| | Urinary tract infection | Urine culture and microscopy | 0 | 0 | 3 | 3 | 36.00 | 50.00 |
| | Vitamin B deficiency | Vitamin $B_{12}$, $B_6$, $B_9$ | 0 | 0 | 6 | 6 | 65.00 | 298.00 |
| | Vitamin D deficiency | Vitamin D | 2 | 4 | 1 | 7 | 35.00 | 175.00 |
| | Vitamin K deficiency | Vitamin K | 0 | 1 | 0 | 1 | 280.00 | |
| **Subgroup 2B (n=45, 9.3% of total)** | | | | | | | | |

Continued

**Table 3** Continued

| Inclusion criteria | Target condition or test purpose | Key analyte or testing method | Mode of test* | | | | Min. cost | Max. cost |
|---|---|---|---|---|---|---|---|---|
| | | | H | S | L | TN | | |
| The testing method used has low sensitivity (limited ability to correctly predict disease) or low specificity (limited ability to correctly rule out disease) or is not reliable as a diagnostic test when used on its own. To reach an accurate diagnosis for the named condition, the gold standard diagnosis, and/or follow-up or parallel tests would be required. | Blood clotting | Prothrombin time | 0 | 0 | 1 | 1 | 55.00 | |
| | Chronic fatigue | Iron, vitamin $B_{12}$, thyroid hormones | 0 | 0 | 1 | 1 | 145.00 | |
| | Coeliac disease | IgA and IgG | 0 | 0 | 4 | 4 | 69.00 | 232.00 |
| | Gastrointestinal candida | IgG anf IgM | 0 | 1 | 0 | 1 | 240.00 | |
| | Giardia | Lambia rapid antigen test | 1 | 0 | 0 | 1 | 106.80 | |
| | Helicobacter pylori | Stool antigen | 1 | 2 | 1 | 4 | 90.00 | 125.00 |
| | Inflammatory bowel disease | Faecal calprotectin | 0 | 1 | 4 | 5 | 75.00 | 95.00 |
| | Kidney disease | Albumin/creatinine ratio dipstick | 2 | 0 | 2 | 4 | 39.00 | 69.60 |
| | Liver disease | Bilirubin dipstick | 1 | 1 | 2 | 4 | 35.99 | 65.00 |
| | Measles, mumps, rubella serology (indicating past infection) | Antibodies | 0 | 0 | 4 | 4 | 49.00 | 90.00 |
| | Pancreatic insufficiency | Faecal pancreatic elastase 1 | 0 | 0 | 1 | 1 | 89.00 | |
| | Pancreatitis | Lipase, amylase | 0 | 0 | 2 | 2 | 59.00 | 62.00 |
| | Urinary tract infection | Urine dipstick (nitrite and leucocyte esterase) | 10 | 0 | 0 | 10 | 25.99 | 215.00 |
| | Vaginal infection | Vaginal pH | 6 | 0 | 0 | 6 | 12.99 | 14.95 |
| Subgroup 2C (n=9, 1.9% of total) | | | | | | | | |
| The test is for a medical condition that has a low pretest probability. Investigating clinically significant disease without clinician guidance or assessment of symptoms and medical history can drive false positive results. | Cytomegalovirus | IgC and IgM antibody | 0 | 0 | 1 | 1 | 59.00 | |
| | Epstein-Barr virus (EBV) | EBV Antibody | 0 | 0 | 1 | 1 | 59.00 | |
| | Glucose-6-Phosphate-Dehydrogenase (G6PD) deficiency | G6PD enzyme assay | 0 | 0 | 1 | 1 | 299.00 | |
| | Lyme disease | Borrelia antibody | 0 | 0 | 5 | 5 | 59.00 | 903.00 |
| | Rotavirus | Rapid antigen test | 1 | 0 | 0 | 1 | 92.40 | |

*Mode of access.
DAPT, direct accessed pathology test; H, home self-tests; L, Lab-collected DAPT; S, self-collected DAPT; TN, total number.

**Table 4** Category 3 results, non-evidence-based commercial 'health checks'

| Inclusion criteria | Target condition or test purpose | Key analyte or testing method | Mode of test* | | | | Min. cost | Max. cost |
|---|---|---|---|---|---|---|---|---|
| | | | H | S | L | TN | | |
| Evaluates normally existing biomarkers or health status in healthy populations; Does not aim at detecting, diagnosing or monitoring of a specific disease or medical condition | Biological age | Glycans | 0 | 1 | 0 | 1 | 599.00 | |
| | Bone health | Serum alkaline, phosphatase, calcium, potassium | 0 | 0 | 2 | 2 | 80.00 | 125.00 |
| | Coenzyme Q10 profile | Coenzyme Q10 level | 0 | 0 | 1 | 1 | 180.00 | |
| | Cytokine profile | Cytokine level | 0 | 0 | 1 | 1 | 425.00 | |
| | Fertility status (male) | Semen analysis | 0 | 0 | 1 | 1 | 75.00 | |
| | Gastrointestinal function | Microbiome analysis | 0 | 11 | 1 | 12 | 109.00 | 1199.00 |
| | Gastrointestinal function | Stool analysis | 0 | 14 | 0 | 14 | 120.00 | 490.00 |
| | Health and wellness profile | Panel consisting of tests for more than one of: full blood count, hormones, organ function, vitamins and minerals | 1 | 0 | 48 | 49 | 18.95 | 749.00 |
| | Hormone profile—female fertility status | Panel consisting of tests for more than one of: Follicle Stimulating Hormone, luteinising hormone, oestradiol, progesterone, testosterone | 0 | 1 | 13 | 14 | 59.00 | 165.00 |
| | Hormone profile—female health | Panel consisting of tests for more than one of: Dehydroepiandrosterone (DHEA), estrone, SHBG, oestradiol, estriol, progesterone, testosterone androstenedione | 0 | 11 | 1 | 12 | 54.00 | 410.00 |
| | Hormone profile—general health | Panel consisting of tests for more than one of: adrenal, growth, thyroid and reproductive hormones | 0 | 10 | 9 | 19 | 59.00 | 239.00 |
| | Hormone profile—male health | Panel consisting of tests for more than one of: DHEA, estrone, oestradiol, testosterone, SHBG androstenedione | 0 | 8 | 3 | 11 | 50.00 | 270.00 |
| | Hormone profile—menopause | Panel consisting of tests for more than one of: oestradiol, estriol, estrone, DHEA | 0 | 2 | 2 | 4 | 81.00 | 179.00 |
| | Hormone profile—ovarian reserve | Anti-Mullerian Hormone | 0 | 0 | 3 | 3 | 98.00 | 299.00 |
| | Hormone profile—sleep quality | Melatonin and cortisol | 0 | 5 | 1 | 6 | 909.00 | 189.00 |
| | Hormone profile—sports all fitness performance | Panel consisting tests for more than one of: Growth and adrenal hormones | 0 | 1 | 14 | 15 | 59.00 | 239.00 |
| | Hormone profile—stress | Salivary cortisol | 0 | 10 | 0 | 10 | 23.00 | 165.00 |
| | Nutrigenomic profile | Genetics | 0 | 2 | 0 | 2 | 249.00 | 550.00 |
| | Nutritional status | Amino acids | 0 | 2 | 0 | 2 | 270.00 | 370.00 |
| | Nutritional status | Ammonia | 0 | 0 | 1 | 1 | 153.00 | |
| | Nutritional status | Chromium | 0 | 0 | 1 | 1 | 108.00 | |

Continued

**Table 4** Continued

| Inclusion criteria | Target condition or test purpose | Key analyte or testing method | Mode of test* | | | | Min. cost | Max. cost |
|---|---|---|---|---|---|---|---|---|
| | | | H | S | L | TN | | |
| | Nutritional status | Copper | 0 | 0 | 2 | **2** | 85.00 | 108.00 |
| | Nutritional status | Essential fatty acids | 0 | 1 | 1 | **2** | 89.00 | 225.00 |
| | Nutritional status | Glutathione | 0 | 0 | 1 | **1** | 335.00 | |
| | Nutritional status | Magnesium | 0 | 0 | 1 | **1** | 138.00 | |
| | Nutritional status | Organic acids | 0 | 8 | 0 | **8** | 74.95 | 597.00 |
| | Nutritional status | Selenium | 0 | 0 | 1 | **1** | 108.00 | |
| | Nutritional status | Iodine | 0 | 1 | 0 | **1** | 155.00 | |
| | Nutritional status | Zinc | 0 | 0 | 2 | **2** | 75.00 | 170.00 |
| | Secretor status (blood typing) | Antigens | 1 | 1 | 0 | **2** | 89.95 | 195.00 |

*Mode of access.
DAPT, direct accessed pathology test; H, home self-tests; L, lab-collected DAPT; S, self-collected DAPT; TN, total number.

or pathological range for a particular individual (the reference interval). Testing asymptomatic and low-risk populations can also sometimes result in the incidental detection of conditions or risk factors that may not benefit from diagnosis or treatment.

### Category 4: Tests whose methods and/or target condition are not recognised by the general medical community (16.7%)

While category 4 tests (table 5) were also non-evidence-based (16.7%), they were condition-specific. We divided these into two subgroups to differentiate how non-evidence-based tests might present different concerns: First, tests that are in themselves non-evidence based, and second, the condition tested for is non-evidence based. For subgroup 4A, without a reliable evidence base to support protocols for diagnosis and treatment, unproven DTC tests can be used to justify treatments for medical conditions that are not clinically recognised,[39] which can lead to further harms. In using tests in subgroup 4B, consumers may be led to believe that they have a condition that requires treatment when there is little evidence to support this. Tests in this subcategory can be used to promote conditions that are not recognised by the medical community and the on-sale of associated non-evidence-based treatments, also known as 'disease mongering'.[40] Most notably, these unproven tests are often high in cost, including the most expensive item found in the review, at $A1947.

### Average cost

The average cost of each category is statistically significantly different (p<0.001). There is a relationship between higher clinical utility and lower cost, and vice versa. Category 1 has the lowest average cost of $A77.16, and smallest range in cost ($A24.95–$A330). Category 2 has more expensive tests with a mean cost of $A138.44 (range of $A13–$A903). Category 3 had an average cost of $A206.08 (range of $A18.95–$A1199) and category 4 had the most expensive tests with an average cost of $A363.11 (range of $A13–$A1947).

### Pretest and post-test consultation and quality assurance

Services such as pretest and post-test consultation are crucial to optimising appropriate test selection and utilisation. However, more than half of all DTC tests (n=273, 56.4%) did not state whether any pretest or post-test counselling was offered (Online supplemental figure 2). Only a quarter of the products (n=123, 25.4%) included a post-test consultation with a healthcare professional, which may include a non-medically trained health practitioner (healthcare professional in Australia is defined as: (a) a medical practitioner, a dentist or any other kind of healthcare worker registered under a law of a state or territory or (b) a biomedical engineer, chiropractor optometrist, orthodontist, osteopath, pharmacist, physiotherapist, podiatrist, prosthetist or rehabilitation engineer. Naturopaths and nutritionists are only health professionals, with regard to the definition above, if they are registered under state or territory law). Another 9.7%

**Table 5** Category 4 results, tests whose methods and/or target condition are not recognised by the general medical community

| Inclusion criteria | Target condition or test purpose | Key analyte or testing method | Mode of test* | | | | Min. cost | Max. cost |
|---|---|---|---|---|---|---|---|---|
| | | | H | S | L | TN | | |
| Subgroup 4A (n=70, 14.5% of total) | | | | | | | | |
| Current evidence does not support the clinical validity of the testing method for the condition it aims to test for. | Environmental toxin exposure | Hair metal and mineral analysis | 0 | 4 | 0 | **4** | 175.00 | 325.00 |
| | Environmental toxin exposure | Mycotoxin test | 0 | 4 | 0 | **4** | 535.00 | 735.00 |
| | Food allergy | Hair analysis | 0 | 6 | 0 | **6** | 29.00 | 297.00 |
| | Food allergy | IgA | 2 | 0 | 0 | **2** | 380.00 | |
| | Food allergy | IgG | 0 | 10 | 12 | **22** | 162.00 | 550.00 |
| | Food sensitivity and intolerance | Antigen leukocyte antibody (ALCAT) | 0 | 13 | 0 | **13** | 325.00 | 1947.00 |
| | Hair loss | Hormones | 0 | 0 | 2 | **2** | 225.00 | 225.00 |
| | Heavy metal exposure | Urine chelation | 0 | 5 | 0 | **5** | 225.00 | 485.00 |
| | Histamine intolerance | Whole blood histamine | 0 | 0 | 3 | **3** | 69.00 | 325.00 |
| | Liver detoxification | Urine metabolites | 0 | 1 | 0 | **1** | 225.00 | |
| | Mental health nutrition (Pfeiffer test) | Zinc, Copper, Zinc-Copper ratio | 0 | 0 | 2 | **2** | 338.00 | 531.00 |
| | Cancer, Autism | Nagalese in blood | 0 | 0 | 1 | **1** | 303.00 | |
| | Mental health disorders | Neurotransmitters in urine | 0 | 1 | 0 | **1** | 360.00 | |
| | Small intestinal bacterial overgrowth | Hydrogen and methane breath test | 0 | 2 | 2 | **4** | 220.00 | 380.00 |
| Subgroup 4B (n=11, 2.3% of total) | | | | | | | | |
| Current evidence does not support the condition tested for as being established and recognised by the general medical community. | Adrenal fatigue | DHEA and Cortisol | 0 | 4 | 1 | **5** | 110.00 | 255.00 |
| | Intestinal permeability (Leaky Gut Syndrome) | Lactulose and mannitol; stool analysis | 0 | 4 | 0 | **4** | 123.00 | 622.80 |
| | Paediatric autoimmune neuropsychiatric disorders syndrome | Antibody | 0 | 0 | 1 | **1** | 1700.00 | |
| | Mental health conditions/Pyrrole disorder | Hydroxyhaemopyrrolin-2-one and neurotoxin | 0 | 1 | 0 | **1** | 199.00 | |

*Mode of access.
DAPT, direct accessed pathology test; H, home self-tests; L, lab-collected DAPT; S, self-collected DAPT; TN, total number.

(n=47) offered a post-test GP telephone consultation to be purchased separately.

Category 1 tests, which were most likely to be clinically useful, were least likely to offer pretest and post-test consultation (Online supplemental figure 3). More than half of home self-tests and self-collected DAPT did not report quality assurance information in the online advertisement. While all lab-collected DAPT were designated for analysis in a NATA-accredited laboratory, there was no detailed information on accreditation status. A very small number of self-collected DAPT were designated for analysis in an overseas laboratory (Online supplemental figure 4).

## DISCUSSION

The wide range and large volume of DTC tests sold online in Australia suggest a sizeable and lucrative market, targeting large numbers of consumers, both with and without symptoms. Accessed in a distinctly different way than for clinician-facilitated tests, there are certain DTC tests with attributes that will provide greater consumer benefits compared with those obtained in mainstream healthcare. The greater availability of DTC tests for stigmatised, undertested and infectious conditions, such as STIs can potentially complement mainstream healthcare by addressing issues of medical underuse.[41] However, only a small proportion of all DTC tests found in this study were of a type likely to provide a clear benefit (category 1, 10.7%). Meanwhile, the majority of tests found in this study would only be offered in mainstream healthcare if there were very clear clinical justifications, if offered at all. These included tests that were recommended against by professional organisations due to concerns of medical overuse, and an array of non-evidenced tests

not recognised by the medical community, as well as tests designed for medical management but rebranded as 'health checks' for asymptomatic people. These types of tests with potential for medical overuse, as identified in our categories of clinical utility, suggest significant implications for harms to consumers and likely unnecessary downstream costs for the healthcare system as a result of further tests and clinical care cascaded by the DTC tests results.

By offering tests not normally obtainable in mainstream clinical care, DTC tests can appear to enhance the convenience and ease of access. People's familiarity with undergoing a pathology test, which are often the same procedures ordered by doctors although under specific clinical circumstances, adds an element of scientific legitimacy to DTC testing.[39] Some products canvassed in the study are overhyped by the innovative appeal of emerging technologies such as nutrigenomic, microbiome and glycan tests, even though these are yet to demonstrate clinical utility in real-world settings.[42 43] In some cases, DTC test results may be used to support diagnoses and treatments that are non-evidence based.[39] This may often target consumers on 'diagnostic odysseys' who are seeking answers for unexplained symptoms or unsatisfactory diagnosis in conventional medicine. For example, a number of tests in category 4 were tests for gastrointestinal conditions, which are known to be difficult to diagnose even in a clinical setting.[8] However, providing non-evidence-based diagnosis and treatments may further harm consumers already in a vulnerable position.

The largest target of DTC tests though are consumers who are healthy and asymptomatic, providing an untapped market for these products.[44] Over 40% of the tests found in this study aim to assess a range of normal, everyday functions such as sleep, ageing, physical activity and hormone levels. The emerging wellness industry relies on the material data from these test results to support a pipeline of justifying and promoting the consumption of nutritional supplements and dietary advice.[45 46] However, testing healthy populations without clear clinical justification will only normalise the notion that people are 'sick until proven healthy'.[14] The downstream consequence of this 'low-value' practice is the triggering of unnecessary medical waste, and more opportunity for incidental and unnecessary diagnoses.

Another concern is the regulatory loophole that allows tests with greater implications for medical overuse to be offered as DAPTs, even though they cannot be supplied in the form of home self-tests in Australia. In a recent review of the *Therapeutic Goods (Excluded Purposes) Specification 2010*, tests for cancer and human genetics were considered too high risk and clinically complex to be used without professional guidance, and therefore, cannot be supplied as DTC home self-tests.[1] Yet our study shows these tests were being sold as lab-collected or self-collected DAPTs. They offer consumers an alternative channel of access to these 'higher risk' tests because they are processed in a laboratory rather than at home. The fact that most DAPTs are processed at NATA accredited laboratories may reinforce their perceived trustworthiness. However, laboratories do not have the capacity to recognise whether the tests are the right ones for the right patient, nor can they influence how consumers make clinical decisions based on test results. Furthermore, our findings suggest there is variation in the quality assurance and the provision of pretest and posttest consultation in DAPTs, and thus compromising the safety and benefits for consumers.

While our findings conclude that the majority of DTC tests currently on the market are unlikely to benefit the everyday consumer, recognisably, some products may be clinically useful if professional consultation and balanced information is provided. For example, there is emerging evidence to show that predictive genetic testing in the general population may lead to clinically useful early interventions if these are clearly available,[47] or have 'personal utility' in guiding advance planning and assisting the coping with uncertain risk status.[48] However, these benefits are facilitated in highly monitored clinical settings, where the assessment of risks, personal circumstances and medical history inform testing decisions, and when extensive professional counselling is provided. The risks and harms of ordering DTC tests, which the TGA review identifies as being particularly relevant for cancers and genetic testing, is that professional guidance and clinical monitoring is lacking. As the balance of harms and benefits of predictive genetic testing is still unclear for many chronic conditions, the call is for an ongoing development of guidelines, professional knowledge and practice in this field, to ensure caution and ethical considerations when offering such tests to asymptomatic, healthy individuals.[47 49] DTC testing may by-pass much of these cautionary processes, which is where the danger lies for consumers. The potential harms of undergoing these DTC tests differ in nuance to those posed by non-evidence-based tests in category 4, however, these products are often sold side-by-side by the same companies, which can mislead consumers.

Our study shows that professional consultation is inconsistently provided, or incurs a separate cost to consumers. Currently, no policy, regulation or industry codes provide clear guidance on the quality and accessibility of this important aspect of testing.

While this study is set in Australia, the growing availability of DTC testing is a topic that is receiving increasing attention internationally.[2 41 50–52] Earlier studies in the USA,[6] UK[7 8] and Europe[53] also show a large variety and volume of DTC tests on sale. What is available to consumers, as reported in this study is time and jurisdiction-specific, due to the different IVD and laboratory regulations that influence the type and manner in which DTC tests can be sold and used.[2 54] Nevertheless, the potential harms to consumers and the healthcare system, and the inadequacy of regulatory frameworks in managing these harms are shared concerns.[17 55] This study has developed a systematic search strategy and an analytical framework

based on clinical utility that can be applied in other jurisdictions to better understand these issues.

To our knowledge, there are no available data in Australia on the extent to which the different types of DTC test are used, nor studies about user motivation or experience. Future research in these directions would inform the design of consumer-centred solutions to address potential harms and provide a better understanding of the social and cultural context of DTC test use.

There are some limitations to the study. The analysis by clinical utility is highly contextual and interpreted by the authors. We acknowledge the potential for the categories to overlap, and that there may be other approaches to analysis. The DTC tests included in this time-specific search may not be exhaustive, as product availability may change over time, and others may be advertised in different platforms or sold in pharmacies. DTC tests offered by private commercial clinics but which are not advertised as a directly purchasable item online, as is often the case for genetic and fertility tests, were also not included. The likelihood is there are many more such tests being sold in the market. This is an overview of products available; data extraction did not include an in-depth qualitative assessment of the textual information in each advertisement. Further study of this aspect would provide insight into the quality and transparency of DTC test advertisement information.

## Conclusion and recommendations

The growing number of DTC tests available for direct purchase online in Australia purport to empower consumers with choice and convenience of access. The categorisation of clinical utility developed in this study shows the majority of currently available products lack clear benefits to consumers. The large offering of tests seldom recommended in a clinical setting may represent an under-recognised driver of medical overuse and consumption of low-value healthcare, particularly among healthy consumers. Unlike publicly funded pathology services and hospital data that are auditable, the harms triggered by DTC tests are not easily tracked and measured. This may mean it remains a relatively hidden problem, with consumers and the healthcare system bearing the brunt of the immediate and long-term costs.

At the policy and regulation level, higher standards of test quality and regulation are needed to support informed decision-making for consumers.[16] Under the new European IVD Regulation there are increasing requirements for IVD companies to provide data to the regulator not just on the quality and analytical performance of tests but also evidence on the clinical performance of the devices, as well as a need for postmarket surveillance on the performance of the tests.[56] Similarly, in Australia, DTC tests, especially those in category 4, should fall under the same scrutiny and regulation before they are allowed to enter the online market. Guidelines or position statements from NATA or the Royal College of Pathologists of Australasia for DTC tests—other than those already issued for genetic tests - would encourage

best practice. Nevertheless, these professional organisation recommendations will be limited to the laboratory end of testing, rather than on consumer-facing marketing and promotion of DTC tests. At the industry level, a commitment to provide balanced information about the evidence base and potential clinical utility for the right populations is needed; as is the assurance of pretesting and post-testing consultation with appropriately qualified professionals. Consumers of varying levels of health literacy must be supported to critically engage and assess the applicability and suitability of different DTC tests. To ensure test results and patient safety are adequately followed up by medical professionals, the introduction of consumer-led reporting systems[41] such as digital registries of test outcomes similar to those used for COVID-19 RATs is recommended. Currently, Choosing Wisely and NICE 'Do Not Do' guidelines are developed for clinical settings where clinicians and consumers are both involved; these could be adapted to support decision-making in the DTC context.

## Author affiliations
[1]School of Health & Society, University of Wollongong, Wollongong, New South Wales, Australia
[2]School of Mathematics and Applied Statistics, University of Wollongong, Wollongong, New South Wales, Australia
[3]School of Public Health, University of Sydney, Sydney, New South Wales, Australia
[4]NSW Health Pathology, Sydney, New South Wales, Australia
[5]University of New South Wales, Kensington, New South Wales, Australia
[6]Sullivan Nicolaides Pathology, Wesley Hospital, Brisbane, Queensland, Australia
[7]Australian Clinical Labs, Victorian Central Laboratory Headquarters, Clayton, Victoria, Australia
[8]Drajon Health, Toodyay, Western Australia, Australia

**Acknowledgements** We would like to thank Tracy Walter and Isabel Serret for providing research assistance with data extraction, and Professor David Handelsman for guidance on sources of information regarding clinical utility of pathology tests used in endocrinology.

**Contributors** PS conceptualised the study and led data collection, with theoretical contributions from KB, SMC and FS. PD provided technical advice and analysis of the quantitative data. ARH, ASJ, DL and MS provided technical advice. All authors contributed to the data analysis and approved the submitted manuscript. PS is responsible for the overall content as guarantor.

**Funding** This study received funding from the Department of Health and Aged Care, Australian Government, National Health and Medical Research Council, grant number #1104136 & #2006545.

**Competing interests** None declared.

**Patient and public involvement** Patients and/or the public were involved in the design, or conduct, or reporting, or dissemination plans of this research. Refer to the Methods section for further details.

**Patient consent for publication** Not applicable.

**Provenance and peer review** Not commissioned; externally peer reviewed.

**Data availability statement** All data relevant to the study are included in the article or uploaded as online supplemental information.

**ORCID iDs**
Patti Shih http://orcid.org/0000-0002-9628-7987
Fiona Stanaway http://orcid.org/0000-0003-2104-3010
Andrew St John http://orcid.org/0000-0001-7564-2823
Katy Bell http://orcid.org/0000-0002-0137-3218

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
