## [Reviewer comments · BMJ Open]

ARTICLE DETAILS

TITLE (PROVISIONAL)	Direct-to-consumer tests advertised online in Australia and their implications for medical overuse: Systematic online review and a typology of clinical utility
AUTHORS	Shih, Patti; Ding, Pauline; Carter, Stacy M; Stanaway, Fiona; Horvath, Andrea; Langguth, Daman; Saad, Mirette; St John, Andrew; Bell, Katy

VERSION 1 – REVIEW

REVIEWER	Jane Tiller Monash University, Public Health Genomics
REVIEW RETURNED	02-May-2023

GENERAL COMMENTS	This paper presents an interesting and thoughtful consideration of the categories of DTC tests being offered in Australia. The paper would benefit from a clearer discussion of DTC tests in the genetics context. Currently there is a broad generalisation of testing in healthy people without any symptoms being of limited clinical utility. While many other tests are offered DTC (as this paper makes clear), one of the primary areas where utility is considered is in DTC genetic tests. As the authors state, this was considered by the TGA when it reviewed its Excluded Purposes Specification recently. The TGA's comments about the tests being too high risk to be supplied without professional guidance, does not mean that the tests are not useful in healthy people. Predictive genetic tests in the general population can have enormous medical benefit to at-risk people without a family or personal history of disease. The fact that these tests should be taken under the care of a health professional to ensure appropriate explanation and risk management advice does not mean that they are not evidence-based or not appropriate for healthy people. A discussion about the difference between genetic results that are not evidence-based and those that have evidence but are inappropriate for DTC provision due to the need for clinical care would assist. This needn't change the categorisation, but the discussion as this investigation pertains to DTC tests would benefit from the clarification.
--

REVIEWER	Claire Duddy University of Oxford, Nuffield Department of Primary Care Health Sciences
REVIEW RETURNED	04-May-2023

GENERAL COMMENTS	This paper provides a valuable snapshot study of the growing consumer market for DTC tests in Australia. The authors provide detailed background information on the need for this study and a
---

	clear exposition of their methods, which are straightforward and appropriate to answer the question. The development of the typology used to categorise DTC tests is explained clearly (and supported by the detail provided in the supplementary file table and figure); this typology may well be of use to other researchers conducting similar studies in other countries. The discussion section provides useful insight into the study findings, providing a balanced perspective on the benefits and potential risks of DTC tests, and providing useful context from other related literatures. The study's limitations are acknowledged, and I agree with the authors' own suggestion that future research examining DTC test advertisements would be a valuable future study. Page 10, line 52 has a typo, I think 'test' here should read as plural 'tests' No statement in relation to funding is provided (or if there is one, I've missed it!)
--	--

REVIEWER	Michelle Kip University of Twente, Health Technology and Services Research
REVIEW RETURNED	08-May-2023

GENERAL COMMENTS	Thank you for providing me the opportunity to review this manuscript. The manuscript shows a very extensive overview on the direct-to-consumer tests which are currently offered in Australia as well as on their clinical utility and implications for medical overuse. The authors provide a clear description on their analysis and on how they decided upon the different categories. In particular the way they classified the tests as having much or little clinical utility is very helpful to judge the usability of these tests. The classifications made are extensively substantiated with literature, and the authors clearly elaborate on the potential limitations of this approach in the discussion. I would suggest to accept the paper after considering a few minor revisions:  - In the search strategy as well as in the paper itself, I was surprised not to see the key word 'point-of-care' being included. I would recommend to explain why you decided not necessary to explicitly include this in your search string. - The paper, and in particular the discussion section, would benefit from some more elaboration on the extent to which these results are comparable to for example the US or Europe. It is already mentioned that the IVD regulation in Europe provides data on for example the analytical performance, but does this for example also affect the number of tests that are commercially available in Australia vs. Europe for example? - Besides 'harms of the test on a population level' (line 54, p.8), could there potentially also be benefits of a test on a population level, for example for contagious diseases? - For the range in costs of the tests as described on line 12, p. 11, I would recommend to also include a median. The costs are maybe highly skewed and only providing a range does not provide insight in this skewness. - In the discussion section, justify whether reviewing the first 200 results of your Google search was expected to be sufficient for not missing tests/products. - It would be very interesting to add some information in the discussion about how often all these different tests are currently being used. This may be very difficult to determine/estimate, but
---

	some insights in that would be very helpful to interpret the implications of the findings of this study.
--	--

VERSION 1 – AUTHOR RESPONSE

Reviewer 1 #1 The paper would benefit from a clearer discussion of DTC tests in the genetics context. Currently there is a broad generalisation of testing in healthy people without any symptoms being of limited clinical utility. While many other tests are offered DTC (as this paper makes clear), one of the primary areas where utility is considered is in DTC genetic tests. As the authors state, this was considered by the TGA when it reviewed its Excluded Purposes Specification recently. The TGA's comments about the tests being too high risk to be supplied without professional guidance, does not mean that the tests are not useful in healthy people. Predictive genetic tests in the general population can have enormous medical benefit to at-risk people without a family or personal history of disease. The fact that these tests should be taken under the care of a health professional to ensure appropriate explanation and risk management advice does not mean that they are not evidence-based or not appropriate for healthy people. A discussion about the difference between genetic results that are not evidence-based and those that have evidence but are inappropriate for DTC provision due to the need for clinical care would assist. This needn't change the categorisation, but the discussion as this investigation pertains to DTC tests would benefit from the clarification.	We have added a section in the discussion to acknowledge the potential benefits of predictive genetic tests for asymptomatic individuals, however, as Reviewer 1 suggests, under a clinically-monitored, professional-guided environment where personal history, context and extensive counselling is provided. We have also ensured these are made distinct from the 'non-evidence-based' tests we described in Category 4. We have pointed to the ongoing debate about the balance of harms and benefits of predictive genetic testing, given that evidence of benefit – whether clinical, social or personal – of predictive genetic tests are still emerging, and will differ from condition to condition. There is ongoing call for development of guidelines, professional knowledge and practice in this field, therefore caution remains paramount in offering such tests to asymptomatic, healthy individuals, particularly those with no family history of disease. DTC testing can by-pass these cautionary processes – which is where the danger lies for consumers.
Reviewer 2 # 1 Page 10, line 52 has a typo, I think 'test' here should read as plural 'tests'	The funding information was provided in the manuscript submission information, but was not displayed in the manuscript version which was

No statement in relation to funding is provided (or if there is one, I've missed it!)	supplied to the reviewers.
Reviewer 3 #1 - In the search strategy as well as in the paper itself, I was surprised not to see the key word 'point-of-care' being included. I would recommend to explain why you decided not necessary to explicitly include this in your search string.	The Royal Australasian College of General Practitioners defines point-of-care testing as “pathology testing performed by, or on behalf of, a medical practitioner at the time of the consultation for diagnosing acute conditions and, to a lesser degree, for monitoring chronic conditions.” https://www.racgp.org.au/advocacy/position-statements/view-all-position-statements/clinical-and-practice-management/point-of-care-testing Point-of-care testing is conducted within a clinical or care-based setting. The person being tested is under the professional care of a healthcare practitioner as a patient. This makes point-of-care testing distinct from the DTC tests we refer to in this article, defined as test that can be directly purchased and initiated by a consumer outside of a clinical setting. We have explicitly clarified this distinction by adding a sentence in the opening paragraph of the introduction where DTC tests are defined.
Reviewer 3 #2 The paper, and in particular the discussion section, would benefit from some more elaboration on the extent to which these results are comparable to for example the US or Europe. It is already mentioned that the IVD regulation in Europe provides data on for example the analytical performance, but does this for example also affect the number of tests that are commercially available in Australia vs. Europe for example?	We have added a sentence in the discussion section about how our results reflect of the globally trend in the rise of DTC tests. As we suggest, the number and type of DTC tests available is time-specific and jurisdiction-specific. The studies we are aware of in US, UK and Europe are therefore less comparable to our current study. Nevertheless, the potential risks and harms to consumers, and the currently inadequate regulatory framework for DTC tests are shared concerns.
Reviewer 3 #3 Besides 'harms of the test on a population level' (line 54, p.8), could there potentially also be benefits of a test on a population level, for example for contagious diseases?	We agree with this comment. We did indeed consider the balance of harms and benefits at a population level when analysing the clinical utility. E.g. this was evident in considering the description and allocation of tests in Category 1. We have adjusted the sentence to ensure the inclusion of ‘benefits’ at a population level, and added ‘infectious’ conditions to the potential benefits of Category 1 tests in the discussion.

Reviewer 3 #4 For the range in costs of the tests as described on line 12, p. 11, I would recommend to also include a median. The costs are maybe highly skewed and only providing a range does not provide insight in this skewness.	Thank you for this recommendation. We have included both the median as well as the mean costs in the main text as well as the abstract.
Reviewer 3 #5 In the discussion section, justify whether reviewing the first 200 results of your Google search was expected to be sufficient for not missing tests/products.	We have added a short section in the Methods section to address this. "Google search engines algorithms display search results by relevance ranking by numerical order: the first ranked results and pages will be the most relevant, with a decline in relevance of search results in subsequent pages. Despite the searches returning large number of total results, only very small, negligible percentages of relevance are expected in later pages, particularly after 200 results. Therefore, we reviewed the first 20 pages (10 results per page, totalling 200 results) from each search. While it can never be guaranteed that this cut-off point will not miss relevant results, it is the most practically and logistically viable approach to conduct the review with a standardised cut-off point for each search scan."
Reviewer 3 #6 It would be very interesting to add some information in the discussion about how often all these different tests are currently being used. This may be very difficult to determine/estimate, but some insights in that would be very helpful to interpret the implications of the findings of this study.	We have addressed this in the discussion section. Currently no data in Australia is available, however, we have made a recommendation for future research in this direction.
Other changes to note	We have made minor editorial adjustments to the text, e.g. corrected typos and other small mistakes. We have added a short acknowledgement section at the end of the manuscript. We have also made the necessary copy edit changes and additional publication statements as requested by the editorial team in an additional email dated 23 June 2023.

VERSION 2 – REVIEW

REVIEWER	Jane Tiller Monash University, Public Health Genomics
REVIEW RETURNED	08-Jul-2023
GENERAL COMMENTS	The reviewer completed the checklist but made no further comments.

REVIEWER	Michelle Kip University of Twente, Health Technology and Services Research
REVIEW RETURNED	03-Jul-2023
GENERAL COMMENTS	Thank you for addressing my comments. I consider your work really interesting and valuable and have no further questions.